# Advanced Glycation End-Products (AGEs) of Lysine and Effects of Anti-TCR/Anti-TNF-α Antibody-Based Therapy in the LEW.1AR1*-iddm* Rat, an Animal Model of Human Type 1 Diabetes

**DOI:** 10.3390/ijms23031541

**Published:** 2022-01-28

**Authors:** Svetlana Baskal, Stefanos A. Tsikas, Olga Begou, Alexander Bollenbach, Sigurd Lenzen, Anne Jörns, Dimitrios Tsikas

**Affiliations:** 1Core Unit Proteomics, Institute of Toxicology, Hannover Medical School, 30623 Hannover, Germany; baskal.svetlana@mh-hannover.de (S.B.); bollenbach.alex@gmail.com (A.B.); 2Dean of Studies Office, Academic Controlling, Hannover Medical School, 30623 Hannover, Germany; tsikas.stefanos@mh-hannover.de; 3Laboratory of Analytical Chemistry, Department of Chemistry, Aristotle University of Thessaloniki, 54124 Thessaloniki, Greece; olina_18@hotmail.com; 4Biomic_Auth, Centre for Bioanalysis and Omics, Centre for Interdisciplinary Research of Aristotle University of Thessaloniki, Innovation Area of Thessaloniki, 57001 Thermi, Greece; 5Institute of Clinical Biochemistry, Hannover Medical School, 30623 Hannover, Germany; lenzen.sigurd@mh-hannover.de (S.L.); joerns.anne@mh-hannover.de (A.J.)

**Keywords:** amino acids, citrullination, diabetes, GC-MS, *N*^ε^-glycation, *N*^ε^-methylation, orthogonal partial least squares discriminant analysis, principal component analysis, RAGE, type 1 diabetes

## Abstract

The LEW.1AR1-*iddm* rat is an animal model of human type 1 diabetes (T1D). Previously, we have shown that combination with anti-TCR/anti-TNF-α antibody-based therapy re-established normoglycemia and increased proteinic arginine-dimethylation in the spleen, yet not in the pancreas. High blood glucose is often associated with elevated formation of advanced glycation end-products (AGEs) which act via their receptor (RAGE). Both anti-TCR and anti-TNF-α are inhibitors of RAGE. The aim of the present work was to investigate potential biochemical changes of anti-TCR/anti-TNF-α therapy in the LEW.1AR1-*iddm* rat. We determined by stable-isotope dilution gas chromatography-mass spectrometry (GC-MS) the content of free and proteinic AGEs and the *N*^ε^-monomethylation of lysine (Lys) residues in proteins of pancreas, kidney, liver, spleen and lymph nodes of normoglycemic control (ngCo, *n* = 6), acute diabetic (acT1D, *n* = 6), chronic diabetic (chT1D, *n* = 4), and cured (cuT1D, *n* = 4) rats after anti-TCR/anti-TNF-α therapy. Analyzed biomarkers included Lys and its metabolites *N*^ε^-carboxymethyl lysine (CML), furosine and *N*^ε^-monomethyl lysine (MML). Other amino acids were also determined. Statistical methods including ANOVA, principal component analysis (PCA) and orthogonal partial least squares discriminant analysis (OPLS-DA) were used to evaluate the effects. Most statistical differences between the study groups were observed for spleen, pancreas and kidney, with liver and lymph nodes showing no such differences. In the pancreas, the groups differed with respect to proteinic furosine (*p* = 0.0289) and free CML (*p* = 0.0023). In the kidneys, the groups differed with respect to proteinic furosine (*p* = 0.0076) and CML (*p* = 0.0270). In the spleen, group differences were found for proteinic furosine (*p* = 0.0114) and free furosine (*p* = 0.0368), as well as for proteinic CML (*p* = 0.0502) and proteinic MML (*p* = 0.0191). The acT1D rats had lower furosine, CML and MML levels in the spleen than the rats in all other groups. This observation corresponds to the lower citrullination levels previously measured in these rats. PCA revealed diametric associations between PC1 and PC2 for spleen (*r* = −0.8271, *p* < 0.0001) compared to pancreas (*r* = 0.5805, *p* = 0.0073) and kidney (*r* = 0.8692, *p* < 0.0001). These findings underscore the importance of the spleen in this animal model of human T1D. OPLS-DA showed that in total sixteen amino acids differed in the experimental groups.

## 1. Introduction

Residues of many proteinogenic amino acids undergo multiple post-translational modifications (PTM). Prominent PTM include the enzymatic citrullination and methylation of the guanidine (*N*^G^) group of arginine (Arg), the enzymatic methylation of the ε-amine (*N*^ε^) group of lysine (Lys) and the non-enzymatic *N*^ε^-glycation of Lys (Figure 1) [1]. Such PTM may have serious consequences for proteins such as: (1) alteration and perhaps possibly entire loss of their naïve inherent functions; (2) acquirement of novel functions; (3) the potential to induce autoimmune diseases; and (4) proteolytic generation of biologically active low-molecular-mass (LMM) substances that exert various biological activities in numerous pathways [1].

The vast majority of Lys methylation occurs in non-histone proteins [2]. Currently, more than 2000 modified Lys-methylations and 1200 different proteins in man and mouse are known [3]. LMM *N*^ε^-methylated Lys metabolites include *N*^ε^-monomethyl lysine (MML) (Figure 1, left panel), *N*^ε^-dimethyl lysine (DML) and *N*^ε^-trimethyl lysine (TML) (not shown). The biological functions of MML, DML and TML are largely unknown. LMM *N*-methylated Arg metabolites are *N*^G^-monomethyl-L-arginine (MMA), *N*^G^,*N*^G^-dimethyl-L-arginine (asymmetric dimethylarginine, ADMA), and *N*^G^,*N*′^G^-dimethyl-L-arginine (symmetric dimethylarginine, SDMA). MMA, ADMA and SDMA are inhibitors of nitric oxide synthase (NOS) activity [4]. The *N*^ε^-glycation of Lys, the *N*^G^-glycation of Arg and the glycation of other amino acid residues in proteins finally yield the LMM advanced glycation end-products (AGEs). Known AGEs of Lys include *N*^ε^-carboxymethyl lysine (CML) (Figure 1, right panel), *N*^ε^-carboxyethyl lysine (CEL), furosine (i.e., [*N*^ε^-(2-furoylmethyl)-L-lysine]) and pentosidine, an AGE composed of Lys and Arg (not shown). LMM *N*^ε^- and *N*^G^-glycated AGEs circulate in the blood and are excreted in the urine of humans and animals [5,6].

AGEs are considered to exert their biological activity via their own receptors of AGEs, i.e., RAGE. RAGE is a multiligand receptor that contributes to the cellular response in diabetic vasculopathy and inflammation [7]. RAGE recognizes families of ligands with diverse structural features. However, to the best of our knowledge, no data are available on the binding affinity of CML and CEL to RAGEs. CML- or CEL-containing heptapeptides were found to have only a low affinity to the V domain of RAGEs of about 100 µM, with the CML- and CEL-free heptapeptides being about six times less affine [8]. RAGE is a multi-ligand cell-bound receptor, which is considered to be highly implicated in pro-inflammatory responses and autoimmunity. Early work found furosine at higher concentrations in nail proteins of diabetic patients compared to non-diabetic humans, with furosine being correlated positively with HbA_1_ [9]. Higher concentrations of AGEs were found in biological fluids in diabetes and other chronic disorders compared to healthy subjects. Based on such findings, AGEs and RAGE are considered to have a causative effect in various inflammatory diseases (reviewed in Refs. [10,11,12,13,14]). Of particular interest in this context is diabetes due to the formation of AGEs from glucose and its metabolites.

Synthetic glycated albumin was found to activate RAGE in mononuclear cells of type 1 diabetes (T1D) and to produce the transcription factor nuclear factor-kappa B (NF-κB) [15]. In an experimental animal model of myasthenia gravis, activation of RAGE by the S100 family of proteins (S100B) was found to exacerbate autoimmune symptoms [16]. RAGE and S100B were detected in the islets of the NOD/*scid* mice that developed diabetes. Blockade of RAGE in islets was found to suppress the autoimmune responses that resulted in T1D in mice [17]. A role for RAGE and AGEs has been suggested for the outcome in heart failure patients with preserved and reduced ejection fraction [18] (see also [6]). Interestingly, the myocardial infarct-exacerbating effect disappeared in RAGE KO and splenectomized C57BL/6 mice [19], suggesting a particular role of the spleen in such processes.

Currently, RAGE and AGEs are the focus of therapy for diabetes’ conditions such as diabetic retinopathy [20], as well as in other inflammatory diseases such as myasthenia gravis [21]. We have shown that the LEW.1AR1-*iddm* rat model of human T1D [22] is a useful experimental tool to study various aspects, including development, progression and therapy of T1D [23,24,25,26,27,28,29]. In a previous study [29], we investigated possible implications of dimethylation and citrullination of Arg residues in proteins in four organs of LEW.1AR1-*iddm* rats under different diabetic metabolic conditions compared to normoglycaemic healthy control rats. This study underlined the importance of *N*^G^-methylation and -citrullination in this model [29]. The aim of the present work was to investigate the effects of anti-TCR/anti-TNF-α therapy in the LEW.1AR1-*iddm* rat on PTM highly specific to glucose metabolism. We expected that the effects of the anti-TCR/anti-TNF-α therapy of the LEW.1AR1-*iddm* rat would be revealed in the tissue concentrations of two prominent AGEs in humans and rats, i.e., *N*^ε^-carboxymethyllysine (CML) and furosine [5,6]. For comparison, we used MML that is not known to act via RAGE (no entry in https://pubmed.ncbi.nlm.nih.gov/?term using the terms “monomethyllysine”, “dimethyllysine” or “trimethyllysine”) (accessed on 15 August 2021). The effects of the combined anti-TCR/anti-TNF-α therapy on additional native and modified amino acids were also investigated in this rat model of human T1D. Results were analyzed by statistical approaches including principal component analysis (PCA) and orthogonal partial least squares discriminant analysis (OPLS-DA).

## 2. Results

### 2.1. Analytical Aspects

For the quantitative measurement of *N*^ε^-glycated and *N*^ε^-methylated Lys derivates developed and validated a gas chromatography-mass spectrometry (GC-MS) method [5]. In the present study, we used this GC-MS method to measure free and proteinic *N*^ε^-glycated and *N*^ε^-methylated metabolites of Lys alongside Lys and other free and proteinic amino acids. We used tissue specimens of pancreas, liver, kidney, spleen and mesenterial lymph nodes of the LEW.1AR1-*iddm* rats, which had been collected and stored frozen at −20 °C in our previous study [29]. Free amino acids and their metabolites were measured without proteolysis (incubation in 6 M HCl, at room temperature (RT) for 20 h). Proteinic amino acids and their metabolites were measured after classical proteolysis (6 M HCl, 110 °C, 20 h). Standard statistical analyses, principal component analysis (PCA) and orthogonal partial least squares discriminant analysis (OPLS-DA) were used to test differences, associations and discriminations between the study groups.

Because the samples of the present study were analyzed about two years after their preparation, first [29] we investigated potential time effects on the amino acid analysis. We compared the previous data (*y*) with the current data from new analyses (*x*) by testing correlation and linear regression analysis for abundant amino acids (i.e., Ala, Thr, Gly, Val, Ser, Leu + Ile, Asp + Asn, Met, Glu + Gln, Orn + Cit, Phe, Tyr, Lys, Arg) and a less abundant amino acid metabolite (i.e., ADMA). All correlations were statistically significant for both the free and the sum of free and proteinic amino acids (*r*-range, 0.69 (Met) to 0.98 (Val)). All regression analyses were statistically significant and linear for the free amino acids (*r*^2^-range, 0.36 to 0.98) and the total (free + proteinic) amino acids (*r*^2^-range, 0.30 to 0.75). The slope values of the regression equations ranged between 0.25 (Met) and 1.34 (Asp + Asn) for the free amino acids, and between 0.46 (Met) and 1.42 (Ala) for the free + proteinic amino acids. There was a relatively small number of discrepancies largely due to the chemical instability of some amino acids such as Met, and due to differences in the calibrators (stable-isotope-labelled amino acids) which had to be freshly prepared. Those amino acids were not considered further for statistical analyses in the present study. All GC-MS analyses were performed quickly under closely similar experimental conditions as described previously [5,29]. The comparison results reported above suggest no considerable changes for free and proteinic amino acids during the storage of the organ tissues over the last two years.

### 2.2. Furosine, N^ε^-Carboxymethyl Lysine (CML) and N^ε^-Monomethyl Lysine (MML)

The tissue weight-corrected content of free and proteinic furosine and CML in the organs of the study rats are summarized in Table 1. The corresponding results for MML are presented in Table 2. The tissue content of free and proteinic furosine, CML and MML differed in the organs and depended upon the kind of treatment. The results are described separately for each organ as follows below.

#### 2.2.1. Pancreas

In the pancreas, there was no difference for free furosine in the normoglycemic control (ngCo) and acute T1D (acT1D) groups. Proteinic furosine differed in the four groups (*p* = 0.0289). Proteinic furosine was about four times lower in the ngCo compared to the other groups. The content of free but not of the proteinic CML (Table 1) and MML (Table 2) differed in the pancreas. Free CML was comparable in the ngCo and acT1D groups, but was lower in the chT1D and cuT1D groups. Free MML differed between the groups, yet the differences were quite small. Thus, with respect to AGEs the results are dichotomic: proteinic furosine is lower, while free CML is higher in the ngCo group.

#### 2.2.2. Kidney

In general, higher tissue contents were measured in the kidney than in the other organs. Free furosine and CML (Table 1) did not differ in the groups. Proteinic furosine and proteinic CML differed in the groups, yet the differences were relatively small. In the kidney, there was no difference for free or proteinic MML in all groups (Table 2).

#### 2.2.3. Liver and Lymph Nodes

The content of the liver and the lymph nodes for furosine, CML and MML was relatively low and there was no difference between the ngCo and the acT1D groups (Table 1 and Table 2).

#### 2.2.4. Spleen

The contents of the spleen of free and proteinic furosine, CML and MML were also relatively low and were in the range 0.05 to 0.4 nmol/mg. In this organ, there were many statistically significant differences between the groups with respect to the biomarkers (Table 1 and Table 2). Previously, we found that spleen might play a particular role in citrullination in the LEW.1AR1-*iddm* rat model of human T1D, as the acT1D group had lower levels of citrullination than the ngCo group [29]. Because of this, we investigated the observations in the spleen in more detail. The results for furosine, CML and MML in rat spleen were examined separately and are shown in Figure 2.

Proteinic furosine (Figure 2A, *p* = 0.0114) differed between the groups. The highest proteinic furosine levels were obtained in the chT1D group. The lower proteinic furosine values in the cuT1D group suggest a decreasing effect of the therapy on the furosine formation. The content of free furosine differed between the ngCo and acT1D groups (Figure 2A, right panel). No statistical differences were found for CML in the spleen (Figure 2B). Proteinic MML differed between the groups (Figure 2C, *p* = 0.0191). Combined treatment with anti-TCR/anti-TNF-α did not reduce the proteinic MML content, suggesting no effect of anti-TCR/anti-TNF-α on the monomethylation of Lys in splenic proteins. The inter-group differences for furosine were larger compared to CML and MML. Like citrullination [29], advanced glycation and monomethylation of Lys seem to be higher in the ngCo group compared to the acT1D group.

Table 3 lists the molar ratio of proteinic to free amino acids in the study groups. Being the precursor of furosine, CML and MML, Lys is of particular importance in the present study. Figure 3 illustrates the molar ratio of proteinic Lys to free Lys in the organs of the study rats. The greatest differences for the Lys molar ratio were observed in the kidney of the rats. In kidney and spleen, the Lys molar ratio was similar in the chT1D and cuT1D groups, as well as in the ngCo and acT1D groups, yet lower in the latter groups. Anti-TCR/anti-TNF-α treatment seems not to have exerted a considerable effect on the Lys molar ratio in the four organs. The data of Table 3 and Figure 3 indicate no appreciable changes in the free and proteinic amino acids including Lys, that could have resulted from altered metabolization and/or inter-organ transfer due to locally changed demands [30,31,32].

### 2.3. Correlations of Amino Acids with Blood Glucose

We tested for correlations between blood glucose and (free as well as proteinic) amino acids in all organs. In the pancreas, proteinic furosine correlated positively with glucose (*r* = 0.4655, *p* = 0.0386). In the kidney, free furosine correlated inversely with blood glucose (*r* = −0.6777, *p* = 0.0010). In the spleen, proteinic CML correlated inversely with glucose (*r* = −0.4558, *p* = 0.0434). In lymph and liver, no correlations were found between blood glucose and furosine, CML or MML. With respect to the D- and L-isomers of free 5-hydroxylysine (i.e., 5-OH-Lys D and 5-OH-Lys L) [5], we did not find correlations in pancreas, kidney, spleen and lymph nodes. In the liver, these isomers correlated to about the same degree: The free acids correlated inversely (*r* = −0.6767, *p* = 0.0185; *r* = −0.7128, *p* = 0.0116), while the proteinic acids correlated positively (*r* = 0.7141, *p* = 0.0113; *r* = 0.7447, *p* = 0.0072). Proteinic 5-OH-Lys D- and L-isomers also correlated with blood glucose in the kidney (*r* = 0.4622, *p* = 0.0402; *r* = 0.5201, *p* = 0.0187), albeit to a lower extent than in the liver.

### 2.4. PCA Results for CML, Furosine, Lysine, Tyrosine and Glucose

Section 2.2 and Section 2.3 have revealed significant differences in the analytes, dependent on the rats’ diabetes condition, and relatively strong pairwise correlations between some of the analytes. In this section, we additionally illustrate by PCA how the analytes relate with each other, and how these relations correspond to the four T1D groups. We employed PCA as a means to reduce dimensionality, because the depiction of multiple variables and treatment/control groups becomes increasingly complex and confusing (see also the Appendix A for more details). PCA has the potential to show if and how acT1D, cuT1D, chT1D or ngCo animals resemble each other, and which aspects may account for similarities and differences. The main results of the PCA approach for pancreas, kidney and spleen are shown in Figure 4. The analyses included glucose, Lys and its AGEs furosine and CML, and Tyr for which no AGEs are known. The results are described separately for each organ as follows below:

#### 2.4.1. Pancreas

Figure 4(A1) shows that blood glucose is not correlated with the other analytes in the pancreas. There is a positive correlation between Tyr, Lys and CML. These analytes exhibit strong negative correlation with furosine (Fu). Figure 4(A1) suggests that Fu seems to be in an antagonistic relationship to CML, Lys, and Tyr. The groups acT1D and chT1D are associated with glucose (Glc), while the groups ngCo and cuT1D form a cluster that is correlated with Tyr, Lys, CML and Fu (Figure 4(B1)). The latter analytes explain the total variability of ngCo and cuT1D. PC2 holds the information that allows discrimination of the groups into three main clusters: (1) acT1D (highest score values); (2) chT1D (middle score values); and (3) ngCo + cuT1D (lowest score values). PC2 for the loading shows that Glc is highly associated to acT1D.

#### 2.4.2. Kidney

The loading plot (Figure 4(A2)) shows a negative correlation between the clusters Tyr + Lys and Glc + CML + Fu. The discrimination of the four groups in the score plot (Figure 4(B2)) is less evident for the kidney than for the pancreas. However, it is undoubtable that PC2 discriminates acT1D from the ngCo + cuT1D groups. The chT1D group is not clearly discriminated.

#### 2.4.3. Spleen

The loading plot (Figure 4(A3)) shows a strong correlation between CML and Fu, which in turn exhibits a lack of association with Glc, Lys and Tyr. There is a strong negative correlation between Glc and Lys, and Glc and Tyr, while Lys and Tyr show some degree of correlation. The score plot for the spleen (Figure 4(B3)) demonstrates that PC2 holds information that discriminates three main clusters: (1) ngCo + cuT1D (highest score values); (2) chT1D (middle score values); and (3) acT1D (lowest) score values. This resembles the situation in the pancreas, albeit in reverse order. Spearman correlation between PC2 and PC1 was negative for the spleen (*r* = −0.8271, *p* < 0.0001), and positive for the pancreas (*r* = 0.5805, *p* = 0.0073) as well as for the kidney (*r* = 0.8692, *p* < 0.0001).

In summary, it is evident from PC2 that for the pancreas and the spleen, Glc is an important variable that contributes to the discrimination process of the study groups.

### 2.5. OPLS-DA Results for Amino Acids

The results of the OPLS-DA analyses are summarized in the Appendix A) and are discussed separately for each organ as follows below.

#### 2.5.1. Pancreas

With respect to the free amino acids of the pancreas, OPLS-DA showed a clear discrimination between the acT1D group and the chT1D and cuT1D groups. In contrast, proteinic amino acids did not show any statistically significant differentiation among the examined groups.

#### 2.5.2. Spleen

In the spleen, free amino acids were only significantly altered between the acT1D group and the chT1D and cuT1D groups. The proteinic amino acid content in the spleen differed in the ngCo and chT1D groups and in the acT1D and chT1D groups (Appendix A).

#### 2.5.3. Kidney

Free amino acids were found to be strongly differentiated again between the acT1D and chT1D groups, and between the acT1D and cuT1D groups. The proteinic amino acids showed more statistically significant differences between the groups.

#### 2.5.4. Liver

No differences were observed between the groups for free or proteinic amino acids.

The molar ratio of proteinic-to-free amino acids differed between the study groups in all examined organs.

## 3. Discussion and Conclusions

T1D is a T cell-mediated autoimmune disease with expression and release of pro-inflammatory cytokines from pancreatic islet infiltrating immune cells. Using the LEW.1AR1-*iddm* rat, an animal model of human T1D [22], we demonstrated that the combination of anti-TCR antibodies with anti-TNF-α antibodies established sustained normoglycemia [26]. This observation strongly argues in favor of a particular role of TCR and TNF-α in the development of T1D. This is of particular interest since TCR and TNF-α are endogenous ligands of RAGE, the receptor of AGEs. RAGE is also expressed by the endocrine cells of the pancreatic islets and is considered to be implicated in the pathogenesis of T1D [33,34]. Additional RAGE ligands include the high-mobility group of box protein 1 (HMGB1), S100 proteins, β-amyloid, β-sheet fibrils and lipopolysaccharide [33]. Research findings gained over the last two decades argue for a link of AGEs and RAGE to the T1D pathogenesis, yet a possible role of AGEs in islet function and survival within homeostasis and disease states is still elusive [28].

Information on the status of *N*^ε^-glycation and *N*^ε^-methylation of Lys in the pancreas and in other anatomically and functionally closely related organs, such as the spleen and the liver, is scarce. In the present work, we investigated a possible implication of a glucose-specific PTM in T1D, i.e., by measuring the *N*^ε^-glycation of Lys residues in tissue proteins in LEW.1AR1-*iddm* rats treated with anti-TCR antibodies and anti-TNF-α antibodies. We measured a wide spectrum of free and proteinic amino acids in specimens of pancreas, spleen, kidney, liver and lymph nodes available from a previous study [29]. In that study, we found that this combination therapy did not alter the *N*^G^-methylation and *N*^G^-citrullination of Arg in the pancreas. However, we found higher *N*^G^-methylation in the cured rats (cuT1D group), notably in the spleen. This observation suggested that the spleen might play a more significant role for the recognition of metabolic changes in T1D than previously generally thought. In contrast, *N*^ε^-methylation of Lys, a glucose-independent PTM, seems not to play a significant role in T1D [29]. We therefore focus our further discussion on the *N*^ε^-glycation of proteinic Lys.

The present study is the first comprehensive animal study on Lys *N*^ε^-glycation and *N*^ε^-monomethylation under several conditions including combined antibody-based therapy of T1D in the LEW.1AR1-*iddm* rat. Free and proteinic CML were found in all organs examined in our study. The highest CML content was found in the kidney of the rats. This observation coincides with the finding from orally administered synthetic ^13^C_2_-CML-containing bovine serum albumin (^13^C_2_-CML-BSA) to mice [35]. These observations may suggest that the kidney may play a key role in the elimination of CML in the urine in rats and mice. The distribution of ^13^C_2_-CML derived from dietary ^13^C_2_-CML-BSA was found not to depend upon the expression of RAGE in the organs [35]. In our study, we did not measure the expression of RAGE in the rat organs. Based on the measurement in various organs of furosine and CML in proteins, the diet-induced obesity (DIO) model of T2D in mice was reported to be useful in studying the effects of glycation in the context of diabetes [36]. Yet, for the sake of completeness, it should be mentioned that in transgenic NOD8.3 mice, exposure to high dietary AGE was found to induce pancreatic β-cell dysfunction, while low dietary AGE was found to decrease pancreatic CML and RAGE in this T1D animal model [37].

Besides T1D, AGEs and RAGEs are assumed to be implicated in T2D. New Zealand Obese (NZO) mice are prone to type 2 diabetes (T2D). Short-term intervention with a carbohydrate-rich diet was found to increase the plasma CML concentration (two-fold) and the RAGE expression (nine-fold) in pancreatic islets in the NZO mice as compared to mice with a carbohydrate-free diet [33].

Our previous and present results indicate that the spleen may play a significant role in T1D. It is assumed that myocardial reperfusion injury (RI) is also triggered by splenic RAGE-dependent mechanisms [19,38,39]. Splenectomy and RAGE-KO mice attained cardioprotection [19,39]. Two independent studies showed that only in the NOD mice, not in other animal models for autoimmune diabetes, did splenectomy influence the T1D incidence [40,41]. The main pro-inflammatory cytokines, especially TNF-α and IL-1β, in the infiltrated pancreatic islets, were mostly derived from immune cells of the pancreas draining into lymph nodes and the spleen. After anti TCR/anti-TNF-α treatment a strong reduction of both cytokines has been observed in the peripheral blood of the successfully treated animals [26]. Independent of the question of the necessity of the spleen during diabetes development, the spleen remained an important source for immune cell reproduction [42].

There appears an emerging role of the spleen in human T1D development, progression and therapy. PTM of certain amino acids, including citrullination, methylation and glycation of Arg and Lys residues in proteins of various organs, can change the inherent protein function. PTM play an important role as an antigen, and induce subsequently autoimmunity that eventually may lead to inflammatory diseases such as rheumatoid arthritis and T1D. PTM are considered important therapeutic targets of diabetes [43]. Measurement of low-molecular-mass PTM metabolites in relevant organs is an important complementary approach in experimental and clinical research.

The serum-free CML concentration in healthy non-diabetic female twins were higher in monozygotic compared to dizygotic and was not associated with the heritability of fasting glucose and HbA_1C_ [44]. Hence, it was hypothesized that largely genetic factors influence the implication of AGEs in vascular disease. Even in T1D, diabetes-independent factors are considered to influence protein glycation [5,45,46]. It is worthy of mention that low-molecular-weight heparin can tightly bind to (*K*_d_, 17 nM) and act as an antagonist to RAGE without the need of glycation thus modulating diabetes in a mouse model of diabetic nephropathy [47]. In this model, glucose and HbA_1c_ concentrations in blood were found to correlate with each other (*r* = 0.89, *p* = 0.0002), with blood proteinic CML being not correlated with glucose or HbA_1c_ concentration [47].

In summary, information on the status of *N*^ε^-glycation and *N*^ε^-methylation of Lys in the pancreas and in other anatomically and functionally closely related organs, such as the spleen and the liver, is relatively scarce. Our present study is the first comprehensive animal study on Lys *N*^ε^-glycation and *N*^ε^-monomethylation under several conditions including combined antibody-based therapy of T1D in the LEW.1AR1-*iddm* rat. The LEW.1AR1-*iddm* rat model of human T1D is useful to study the effects of main PTM including citrullination, methylation and glycation of proteins in pancreas, kidney, liver, spleen and lymph nodes in different diabetes conditions. Our studies confirm a particular yet not fully understood role of the spleen in the evolvement and pharmacotherapy of T1D. The presumably very low or even absent affinity of free, non-proteinic AGEs CML, CEL and furosine to RAGE [8] suggests that these AGEs in tissue, blood and urine [5,6,48] may serve as biomarkers of glycation rather than as activators of RAGE. A limitation of our study is the lack of RAGE expression measurements in the LEW.1AR1-*iddm* rat organs. Such analyses may help elucidating underlying mechanisms in physiology and pharmacotherapy of RAGE in this model.

## 4. Materials and Methods

### 4.1. Experimental Groups of the LEW.1AR1-iddm Model of Human T1D

In the present work, we analyzed free and proteinic amino acids and metabolites in homogenates prepared in a previous study [29]. We analyzed specimens of organs of LEW.1AR1-*iddm* rats harvested in a previous, described in detail study [26]. The Congenic LEW.1AR1-*iddm* (IDDM) rats (for details see http://www.mh-hannover.de/34926.html) (accessed on 24 November 2021) were generated and maintained as described elsewhere [22,23,24,25,26]. Experimental procedures were approved by the District Government of Hannover (LAVES, No 33-42502-05/958 and 33.9-42502-04/16/2115) in accordance with the guidelines for the care and use of laboratory animals.

Group 1 (ngCo) comprised six (three males, three females) healthy, normoglycaemic control rats (blood glucose, 5.1 ± 0.3 mM) with an age of 120 days without therapy. Group 2 (acT1D) comprised six (three males, three females) only with IgG-treated acute diabetic IDDM rats (diabetes onset, 61.8 ± 4.0 days; blood glucose, 22.9 ± 2.7 mM). Pancreatic islets of the normoglycaecemic control rats showed a high amount of pancreatic beta cells surrounded by the other islet cells without any sign of immune cell infiltration [22,26]. Group 3 (chT1D) represented four (two males, two females) chronic diabetic IDDM rats (blood glucose, 14.1 ± 1.2 mM) [22], which were treated with rat-specific insulin pellets (Lin Shin Canada Ltd., Toronto, Canada) over six months. After this long period after diabetes manifestation, the number of pancreatic islets was markedly reduced showing pancreatic islets mostly without beta cells and in parallel without infiltrating immune cells because of the complete loss of beta cells.

Group 4 (cured rats, cuT1D) comprised four (four males) originally acute diabetic IDDM rats, which were treated for 5 consecutive days immediately after disease manifestation. The combination therapy consisted of a rat-specific anti-TCR antibody (Clone: R73; Bio-Rad, Munich, Germany; intravenous administration of 0.5 mg/kg bodyweight) and a rat-specific anti-TNF-α antibody (Janssen Research & Development, Spring House, PA; intravenous administration of 5 mg/kg body weight). The rats of group 4 returned to normoglycaemia during therapy (blood glucose, 5.9 ± 0.4 mM) which lasted for 60 days after the end of therapy. The pancreatic islets of the cuT1D group resembled beta cells in the same amount in the islets as those of the normoglycaemic control group (ngCo) without any signs of immune cell infiltration as previously reported [26]. Evidence of manifestation and successful therapy of T1D was reported in detail in previous study by means of different techniques besides measurement of blood glucose [26]. These experimental groups were previously characterized for citrullination [29].

### 4.2. Organ Homogenization and GC-MS Measurement of Tissue Free and Proteinic Amino Acids and Metabolites

Frozen tissue (−80 °C) from pancreas, kidney, liver, spleen and mesenterial lymph nodes specimens were obtained from two animal studies comprising each two experimental groups as detailed above. The groups were analyzed for pancreatic characteristics [26]. Thawed organ pieces (55 ± 32 mg for both studies) were homogenized in 1-mL aliquots of ice-cooled phosphate buffered (67 mM, pH 7.4) saline containing cOmplete™ Mini EDTA-free Protease Inhibitor Cocktail (Sigma-Aldrich, Germany) resulting in tissue protein concentration of 5.9 ± 3.7 mg/mL. Amino acids had been measured in two 10-µL aliquots of each homogenate sample without (20 °C, 20 h) and with proteolysis (110 °C, 20 h) with 6 M HCl [5,29]. Remaining homogenate samples were frozen at −18 °C until repeated analysis. Samples were thawed in an ice bath immediately before hydrolysis. Two 10-µL aliquots of thoroughly vortexed samples were taken and incubated with 6 M HCl for 20 h at room temperatures for the free analytes and at 110 °C for the sum of free and proteinic analytes in tightly closed autosampler glass vials. Amino acids, PTM metabolites and AGEs in the homogenate samples of all available organs were measured by GC-MS after esterification with 2 M HCl in methanol and amidation with pentafluoropropionic anhydride in ethyl acetate as described recently [5]. The content of free and proteinic amino acids in the homogenates was corrected for the weight of the tissue and is reported as nmol analyte per mg tissue weight (nmol/mg).

### 4.3. Statistical Analysis and Data Presentation

Data analyses were performed using GraphPad Prism 6 for Windows (GraphPad Software, San Diego, CA, USA). Data are presented as mean ± SD or mean ± SEM. Comparison between groups was performed using unpaired *t*-test. Comparison between organs was performed using one-way ANOVA with Tukey’s multiple comparisons test. Spearman and in few cases Pearson correlation analyses were used to test for associations as appropriate. A two-sided *p*-value of less than 0.05 was considered statistically significant.

PCA was performed with STATA 14 (StataCorp, College Station, TX, USA) for glucose, CML, furosine, Lys and Tyr (see main document) and with SIMCA 13.0.2 (UMETRICS AB Sweden) in unit variance (UV) scaling for all amino acids (see Appendix A). Data were further analyzed with PLS-DA, OPLS and OPLS-DA. The validity of the obtained models was assessed using the cross-validation parameters (R2X: R2Y and Q2Y), in combination with loadings, permutation and VIP plots and the *p*-value of cross-validated analysis of variance (CV-ANOVA). Only features with VIP value > 1 were considered statistically significant. In order to find any metabolic differences between the studied groups univariate statistical analysis using two-tailed *t*-test, with unequal variance algorithm (a threshold of *p*-value or *q*-value (FDR corrected) was set at 0.05) was performed to check the impact of each metabolite on the tested hypothesis.

## Figures and Tables

**Figure 1 ijms-23-01541-f001:**
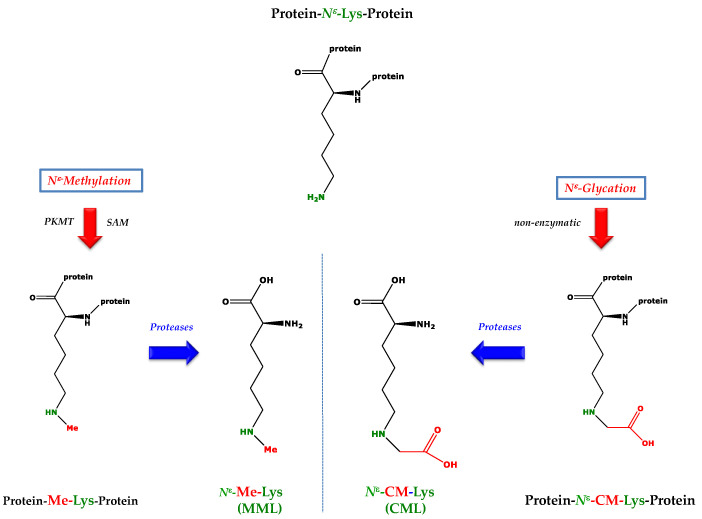
Simplified schematic of two post-translational modifications of lysine. (**Left**) The *N*^ε^-monomethylation of Lys is catalyzed by protein-lysine methyl transferases (PKMT; EC 2.1.1.43). Proteolysis of the *N*^ε^-monomethylated Lys protein releases *N*^ε^-monomethyl lysine (MML). (**Right**) The *N*^ε^-glycation of Lys is non-enzymatic. The advanced glycation end-product of the reaction of *N*^ε^-Lys with glyoxal (O=CH-CH=O) is *N*^ε^-carboxymethyllysine (CML). Abbreviations: CM, carboxymethyl; Me, methyl; SAM, *S*-adenosylmethionine; SAM is the cofactor of PKMT.

**Figure 2 ijms-23-01541-f002:**
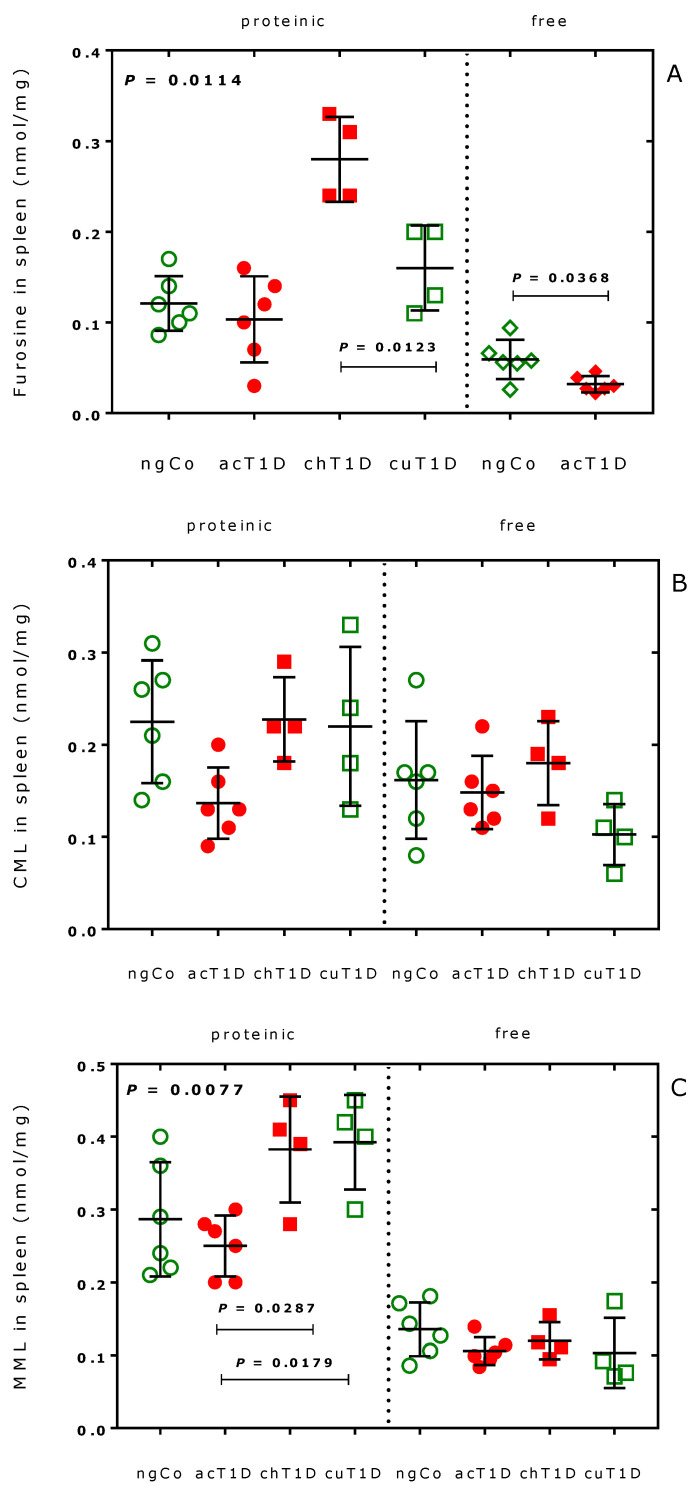
Content (nmol/mg spleen) of proteinic and free (**A**) furosine; (**B**) carboxymethyl lysine (CML); and (**C**) monomethyl lysine (MML) in the spleen of the four rat groups. Data are shown as mean with standard deviation. Statistical significance was tested by one-way ANOVA and Tukey’s multiple comparisons test. ngCo, normoglycaemic control; acT1D, acute type 1 diabetes; chT1D, chronic type 1 diabetes; cuT1D, cured type 1 diabetes (cuT1D) rats. See also Table 1 and Table 2.

**Figure 3 ijms-23-01541-f003:**
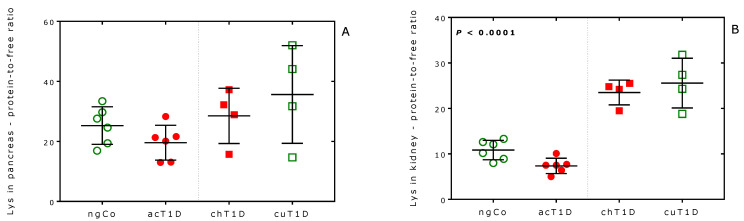
Molar ratio of proteinic to free lysine (Lys) in the four rat groups in (**A**) pancreas; (**B**) kidney; (**C**) spleen; and (**D**) lymph nodes and liver. Data are shown as mean with standard deviation. One-way ANOVA and Tukey’s multiple comparisons test in (**B**,**C**). ngCo, normoglycaemic control; acT1D, acute type 1 diabetes; chT1D, chronic type 1 diabetes; cuT1D, cured type 1 diabetes rats. *p* values in (**D**) were obtained by unpaired *t*-test.

**Figure 4 ijms-23-01541-f004:**
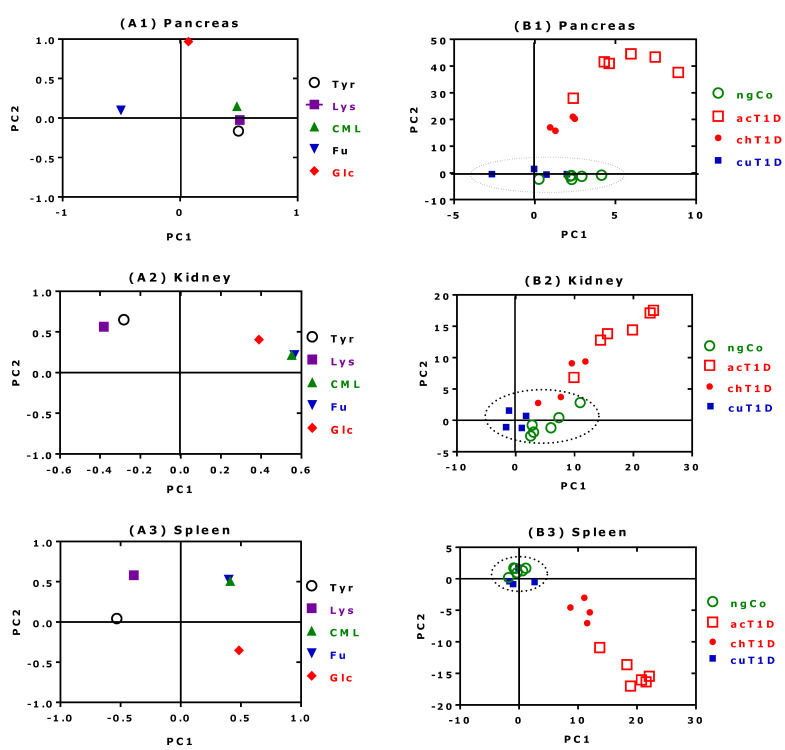
PCA results for pancreas (**A1**,**B1**); kidney (**A2**,**B2**); spleen (**A3**,**B3**). Left: plot of analytes‘ loadings in the PC1/PC2 plane, defined on an interval from (−1;1) for glucose (Glc); furosine (Fu); lysine (Lys); tyrosine (Tyr); and CML. Right: plot of component scores in the PC1/PC2 plane. ngCo, normoglycaemic control; acT1D, acute type 1 diabetes; chT1D, chronic type 1 diabetes; cuT1D, cured type 1 diabetes rats. The Principal Components’ Eigenvalues and the explained variances are found in the Appendix A.

**Table 1 ijms-23-01541-t001:** Content (mean ± SEM; nmol/mg tissue) of free and proteinic furosine and carboxymethyllysine in organ tissues of normoglycemic control (ngCo); acute type 1 diabetes (acT1D); chronic type 1 diabetes (chT1D); and cured type 1 diabetes (cuT1D) rats.

Group	Free	Proteinic	Free	Proteinic
	Furosine		Carboxymethyllysine	
PANCREAS				
ngCo	0.0412 ± 0.0094	0.0206 ± 0.0089	0.6033 ± 0.0092	0.1200 ± 0.0656
acT1D	0.0410 ± 0.0043	0.0743 ± 0.0183	0.7700 ± 0.0856	0.1250 ± 0.0296
chT1D	n.d.	0.0883 ± 0.0102	0.2800 ± 0.0212	0.0550 ± 0.0119
cuT1D	n.d.	0.0825 ± 0.0168	0.2525 ± 0.0364	0.0625 ± 0.0225
	n.s.	*p* = 0.0289	*p* = 0.0023	n.s.
KIDNEY				
ngCo	0.2022 ± 0.0489	0.5583 ± 0.0260	0.885 ± 0.1356	0.3283 ± 0.0456
acT1D	0.0708 ± 0.0118	0.4950 ± 0.0796	0.5483 ± 0.1269	0.1950 ± 0.0273
chT1D	0.0863 ± 0.0209	0.5175 ± 0.0984	0.4175 ± 0.0978	0.215 ± 0.0444
cuT1D	0.1410 ± 0.0146	0.5450 ± 0.0491	0.4375 ± 0.0379	0.1600 ± 0.0234
	n.s.	*p* = 0.0076	n.s.	*p* = 0.0270
LIVER				
ngCo	0.0193 ± 0.0030	0.1500 ± 0.0146	0.0752 ± 0.0083	0.1288 ± 0.0146
acT1D	0.0165 ± 0.0010	0.1800 ± 0.0159	0.0727 ± 0.0073	0.1362 ± 0.0112
	n.s.	n.s.	n.s.	n.s.
SPLEEN				
ngCo	0.0592 ± 0.0089	0.1210 ± 0.0123	0.1617 ± 0.0260	0.2250 ± 0.0272
acT1D	0.0318 ± 0.0036	0.1033 ± 0.0194	0.1483 ± 0.0162	0.1367 ± 0.0158
chT1D	n.d.	0.2800 ± 0.0235	0.1800 ± 0.0227	0.2275 ± 0.0229
cuT1D	n.d.	0.1600 ± 0.0235	0.1025 ± 0.0165	0.2250 ± 0.0272
	*p* = 0.0368	*p* = 0.0114	n.s.	n.s.
LYMPH NODE				
ngCo	0.0313 ± 0.0107	0.0703 ± 0.0432	0.1010 ± 0.0149	0.0808 ± 0.0390
acT1D	0.0140 ± 0.0026	0.0252 ± 0.0150	0.0892 ± 0.0102	0.0276 ± 0.0086
	n.s.	n.s.	n.s.	n.s.

Abbreviations. n.d., not detected; n.s., not significant.

**Table 2 ijms-23-01541-t002:** Content (mean ± SEM; nmol/mg) of free and proteinic monomethyllysine (MML) in the organ tissues of the normoglycemic control (ngCo); acute type 1 diabetes (acT1D); chronic type 1 diabetes (chT1D); and cured type 1 diabetes (cuT1D) rats.

Group	*N*^ε^-Monomethyllysine (MML)	
	Free	Proteinic
PANCREAS		
ngCo	0.0897 ± 0.0215	0.0385 ± 0.0108
acT1D	0.1183 ± 0.0168	0.1370 ± 0.0131
chT1D	0.0755 ± 0.0114	0.1295 ± 0.0231
cuT1D	0.1038 ± 0.0118	0.1538 ± 0.0303
	*p* = 0.0014	n.s.
KIDNEY		
ngCo	0.1123 ± 0.0322	0.1733 ± 0.0353
acT1D	0.0988 ± 0.0146	0.1238 ± 0.0159
chT1D	0.0803 ± 0.0109	0.1715 ± 0.0404
cuT1D	0.0755 ± 0.0029	0.1263 ± 0.0175
	n.s.	n.s.
LIVER		
ngCo	0.0413 ± 0.0026	0.0630 ± 0.0039
acT1D	0.0323 ± 0.0037	0.0638 ± 0.0075
	n.s.	n.s.
SPLEEN		
ngCo	0.1357 ± 0.0150	0.2867 ± 0.0320
acT1D	0.1058 ± 0.0077	0.2500 ± 0.0171
chT1D	0.1198 ± 0.0127	0.3825 ± 0.0364
cuT1D	0.1033 ± 0.024	0.3925 ± 0.0325
	n.s.	*p* = 0.0191
LYMPH		
ngCo	0.0858 ± 0.0198	0.3633 ± 0.1016
acT1D	0.059 ± 0.0083	0.1983 ± 0.0736
	n.s.	n.s.

Abbreviations. n.s., not significant.

**Table 3 ijms-23-01541-t003:** Molar ratio (mean ± SEM; nmol/mg) of proteinic to free amino acids in the organ tissues of the normoglycaemic control (ngCo); acute type 1 diabetes (acT1D); chronic type 1 diabetes (chT1D); and cured T1 diabetes (cuT1D) rats.

	Phe	Tyr	Lys	Arg	Trp	Ala	Thr	Gly	Val	Ser	Leu/ Ile ^a^	Asp/ Asn ^a^	Pro	Met	Glu/ Gln ^a^	Orn/ Cit ^a^
PANCREAS																
ngCo	28.5	15.5	25.6	35.7	6.0	12.6	30.2	5.4	84.6	12.5	41.7	36.6	24.6	3.22	8.75	0.84
acT1D	25.5	11.8	19.6	27.0	6.5	9.6	21.9	4.5	52.9	10.9	33.4	27.8	19.7	2.70	6.6	0.59
chT1D	25.8	16.8	28.5	29.2	5.0	8.6	47.2	5.0	63.8	10.9	36.9	21.9	27.5	3.32	8.7	0.98
cuT1D	21.8	12.8	35.6	38.6	3.5	7.7	25.4	5.2	85.1	12.1	36.3	25.2	25.4	3.3	8.7	1.38
KIDNEY																
ngCo	11.7	5.1	10.9	11.3	4.0	4.5	8.7	2.8	12.6	4.3	13.1	10.1	9.4	2.8	6.3	0.46
acT1D	8.9	4.4	7.4	8.7	3.9	3.5	5.9	2.6	8.4	3.1	9.1	7.1	7.1	2.5	5.2	0.33
chT1D	17.4	9.0	23.5	18.7	5.5	8.5	15.3	3.7	26.4	5.8	22.6	17.7	12.2	2.9	7.2	1.3
cuT1D	19.7	9.8	25.6	22.1	3.4	9.7	16.9	3.9	30.8	6.6	25.4	18.6	12.5	3.3	8.9	1.5
LIVER																
ngCo	37.2	18.7	32.2	64.3	8.5	8.7	37.6	5.7	75.2	15.1	46.3	70.8	54.0	5.4	13.7	1.9
acT1D	34.8	17.0	28.3	66.5	6.3	9.9	34.5	5.5	59.0	13.0	42.0	54.7	57.8	5.1	11.1	1.0
SPLEEN																
ngCo	12.6	4.3	18.2	11.6	3.3	9.2	13.7	3.6	19.0	6.8	17.1	8.3	13.7	2.8	5.8	2.7
acT1D	13.2	4.2	19.1	11.2	3.5	8.4	12.3	3.6	19.6	6.3	17.3	9.8	13.3	2.4	5.2	2.3
chT1D	16.9	7.3	26.5	18.4	3.9	9.9	19.6	4.2	29.1	7.1	22.2	16.1	18.5	3.3	7.9	3.2
cuT1D	15.6	5.1	28.4	19.2	5.7	10.2	18.7	4.3	30.3	7.9	22.2	13.5	18.0	2.8	6.9	4.0
LYMPH																
ngCo	6.1	2.6	10.1	9.01	3.2	5.6	9.5	3.4	12.4	4.9	10.5	6.0	10.3	2.4	5.2	2.8
acT1D	6.2	2.5	8.2	7.1	2.5	4.4	7.2	2.8	10.8	3.9	9.9	5.5	8.8	2.0	3.8	1.5

^a^ for methodological reasons [3], the sum of Leu and Ile, Asp and Asn, Glu and Gln, and ornithine (Orn) and citrulline (Cit) was determined by GC-MS and reported.

## Data Availability

The study did not report any data.

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
