# Peer review of "Advanced Glycation End-Products (AGEs) of Lysine and Effects of Anti-TCR/Anti-TNF-α Antibody-Based Therapy in the LEW.1AR1-iddm Rat, an Animal Model of Human Type 1 Diabetes"

_ijms, 2022, doi:10.3390/ijms23031541_

Round 1
Reviewer 1 Report
Subsequent reports found higher concentrations of AGEs in biological fluids in diabetes and other chronic disorders compared to healthy subjects. Based on such findings, RAGE is considered to have a causative effect in various inflammatory diseases.
Cite papers
Figure 2: Legends need to be elaborated more. Please describe the groups properly in legends
Figure 3: Please elaborate on the legends.
Also, Fig3 A-B, please repeat the measurement for the last group (cuT1D). The standard deviation is high.
Figure 4: Please mention A1, B1, etc in the legends.
Why did the authors have so less analytes in A1-A3?
Author Response
see PDF

Reviewer 2 Report
Although the detailed analyses of AGEs of lysine in anti-TCR/anti-TNF-α-based are interesting, numbers of points need clarifying and certain statements require further justification. These are given below.
<Points>
- The authors described, “The studies were approved by the local Institutional Ethics Committees on Animal Care and Experimentation.” (lines 360-361) without showing the approval number(s) and date. The authors should show the approval number(s) and date.
- Supplementary data (Tables S1-S6 & Figures S1-S6) should be changed to regular Tables and Figures because there is no limitation in we-based journals such as IJMS and the supplementary data seem to be important for this study.
- Concerning the analyses of pancreas, it is far better to add some data of how islets were. Although the authors reported how islets were in LEW.1AR1-iddm rats (Diabetes 64, 2880-2891, 2015), there is no classification of acT1D, chT1D, and cuT1D in the previous work.
- As RAGE is receptor for AGEs and the expression of RAGE is changed by glucose and RAGE ligands such as AGEs, S100, HMGB1, and LPS, it would be better to add RAGE expression (preferably RAGE, sRAGE, and esRAGE).
Author Response
see PDF

Round 2
Reviewer 2 Report
Although the authors revised appropriately, some points should be modified. These are given below.
<Points>
- Figure 3 in the REVISED VERSION is dark background figure. For the benefit of readers, the background should be changed to white like Figures 2 and 4 in the REVISED VERSION.
- Line 7: The authors changed from “, Germany” to “and Germany” in the manuscript. However, it is still “, Germany” in the supplementary. The authors should keep consistent.
Concerning animal experiments, the authors appropriately revised in the text of REVISED VERSION. However, it was not precisely indicated in the RESPONSE: The description appears in lines 420-422 but not 450-452, 456-458, 461-464, 471-474 in the REVISED VERSION.
Author Response
- Figure 3 in the REVISED VERSION is dark background figure. For the benefit of readers, the background should be changed to white like Figures 2 and 4 in the REVISED VERSION.RESPONSE: in our files, the background of Figure is white.
- Line 7: The authors changed from “, Germany” to “and Germany” in the manuscript. However, it is still “, Germany” in the supplementary. The authors should keep consistent.RESPONSE: in our files, we did not find "and Germany"
Concerning animal experiments, the authors appropriately revised in the text of REVISED VERSION. However, it was not precisely indicated in the RESPONSE: The description appears in lines 420-422 but not 450-452, 456-458, 461-464, 471-474 in the REVISED VERSION.
RESPONSE: The Reviewer is right. We apologize for our mistake